# Comparative Analysis of Enzymatic Transglycosylation Using *E.* *coli* Nucleoside Phosphorylases: A Synthetic Concept for the Preparation of Purine Modified 2′-Deoxyribonucleosides from Ribonucleosides

**DOI:** 10.3390/ijms23052795

**Published:** 2022-03-03

**Authors:** Mikhail S. Drenichev, Vladimir E. Oslovsky, Anastasia A. Zenchenko, Claudia V. Danilova, Mikhail A. Varga, Roman S. Esipov, Dmitry D. Lykoshin, Cyril S. Alexeev

**Affiliations:** 1Engelhardt Institute of Molecular Biology, Russian Academy of Sciences, Vavilova Str. 32, 119991 Moscow, Russia; mdrenichev@mail.ru (M.S.D.); vladimiroslovsky@gmail.com (V.E.O.); kolomatchenkoa@yandex.ru (A.A.Z.); yourwintersunrise@gmail.com (C.V.D.); misha.varga.00@bk.ru (M.A.V.); 2Laboratory of Biopharmaceutical Technologies, Shemyakin-Ovchinnikov Institute of Bioorganic Chemistry, Russian Academy of Sciences, Ulitsa Miklukho-Maklaya, 16/10, GSP-7, 117997 Moscow, Russia; refolding@mail.ru (R.S.E.); ldd-94@yandex.ru (D.D.L.)

**Keywords:** 7-methyl-2′deoxyguanosine, nucleoside phosphorylase, enzymes, transglycosylation, biologically active nucleosides, fluorine, benzyladenine, kinetin

## Abstract

A comparative analysis of the transglycosylation conditions catalyzed by *E. coli* nucleoside phosphorylases, leading to the formation of 2′-deoxynucleosides, was performed. We demonstrated that maximal yields of 2′-deoxynucleosides, especially modified, can be achieved under small excess of glycosyl-donor (7-methyl-2′-deoxyguanosine, thymidine) and a 4-fold lack of phosphate. A phosphate concentration less than equimolar one allows using only a slight excess of the carbohydrate residue donor nucleoside to increase the reaction’s output. A three-step methodology was elaborated for the preparative synthesis of purine-modified 2′-deoxyribonucleosides, starting from the corresponding ribonucleosides.

## 1. Introduction

Nucleoside-based drugs are widely used in clinical practice: half of the antiviral medications and a quarter of antitumor ones belong to this group of natural compounds [1,2]. 2′-Deoxynucleosides represent synthetic substrates for wide application in medicinal chemistry and pharmacology. Several cytostatic nucleoside-based drugs commonly used in medical practice (cladribine, pentostatin, gemcitabine, clofarabine) consist of heterocyclic base joined to 2-deoxyribose residue by β-*D*-glycoside bond [3,4]. Due to less chemical stability of *N*-glycosidic bond in deoxyribonucleosides, their preparation is associated with a range of difficulties, making the latter commercially less available products than ribonucleosides. Nucleoside derivatives can be synthesized by chemical and enzymatic methods. Chemical methods include modifying initial natural nucleosides, either obtaining heterocyclic bases and/or monosaccharides derivatives with subsequent condensation forming target nucleosides. In this case, the key synthesis stage is forming an *N*-glycoside bond. To date, convenient and efficient chemical methods for the production of ribonucleosides by glycosylation of modified heterocyclic base with an excess of peracylated glycosyl-donor in the presence of Lewis acids have been developed. Still, the yields of β-*D*-2′-deoxyribonucleosides in the same conditions are significantly lower because of the formation of a mixture of α- and β-anomeric products [5,6]. In some cases, this method increases the yield of the target product to 50–67% [5,7,8,9]. Chemical transglycosylation proceeds under treatment of protected purine or pyrimidine nucleoside and pyrimidine or purine base respectively in the presence of Lewis acid (TMSOTf, SnCl_4_), but also leads to the formation of α- and β-anomeric mixture when 2′-deoxyribonucleosides act as initial compounds. Therefore, increasing the selectivity of the β-deoxyribonucleosides formation over α-anomers requires additional substrates and chemical synthesis steps, which decrease the overall yield of target products. Enzymatic methods of *N*-glycoside bond formation complement chemical ones and, in some cases, have undoubted advantages, such as regio- and stereoselectivity [3,4,5]. Nucleoside phosphorylases (NPs) are widely used to produce practically important natural nucleosides and their analogues. NPs play a vital role in the metabolism of nucleosides. The most specific enzyme among NPs is thymidine phosphorylase (TP) (EC 2.4.2.4), whose substrates are thymidine and 2′-deoxyuridine [3,4]. Uridine phosphorylase (UP) (EC 2.4.2.3), in addition to these nucleosides, also performs phosphorolysis of uridine (Figure 1, step 1). Bacterial purine nucleoside phosphorylases (PNPs) are less specific (EC 2.4.2.1), converting both ribo- and 2′-deoxy purine nucleosides as substrates. They are widely used to obtain nucleoside-based drugs and their analogues [1,2,3,4,10]. Methods of enzymatic transglycosylation represent the transfer reaction of a carbohydrate residue from one heterocyclic base to another.These reactions are catalyzed by NPs, which carry out reversible phosphorolysis of ribonucleosides/2′-deoxyribonucleosides forming the corresponding heterocyclic base and α-*D*-(2-deoxy) ribofuranose-1-phosphate ((d)Rib-1-P). The equilibrium of the phosphorolysis reaction is shifted towards the formation of nucleosides, more significantly in the case of purine ones [1,3,4,11,12], which allows using two conjugated phosphorolysis reactions of nucleoside-donor and a nucleoside, containing heterocyclic base-acceptor to conduct an enzymatic transglycosylation reaction, during which a carbohydrate residue is transferred from a pyrimidine or purine nucleoside donor to a purine heterocyclic base-acceptor (Figure 1).

This general procedure makes it possible to obtain new modified nucleosides depending on a set of starting compounds and the substrate specificity of NP. Various approaches for synthesizing modified nucleosides using an enzymatic phosphorolysis of 7-methyl-2′-deoxyguanosine (7-Me-dGuo) to form α-*D*-2-deoxyribose-1-phosphate (dRib-1-P) as carbohydrate residue donor were previously described [13,14,15,16]. Then, our research has proposed a mathematical model of the transglycosylation process, which can be used to quantify the effect of initial conditions on the yield of transglycosylation reaction [17]. This work represents a continuation of our previous investigations. In this paper, we discuss the influence of the reagents’ ratio and the role of the phosphate-anion concentration on yields of transglycosylation reaction for the synthesis of 2′-deoxyribonucleosides that can act as active pharmaceutical ingredients. Based on these results, we propose the three-step procedure for the enzymatic synthesis of several purine β-*D*-2′-deoxyribonucleosides from corresponding β-*D*-ribonucleosides containing chemically modified heterocyclic base.

## 2. Results

Transglycosylation reaction catalyzed by NPs represents the most widely used method for biochemical synthesis of nucleosides, in which a carbohydrate residue is transferred from a nucleoside (Nuc1) to a heterocyclic base (Base2), forming a new nucleoside (Nuc2) (Figure 1). According to Figure 1, transglycosylation reaction proceeds through the formation of Rib-1-P or dRib-1-P. The synthesis of purine 2′-deoxynucleosides from pyrimidine ones and *vice versa* required the participation of two enzymes of NPs’ family: PNP and TP. Therefore, we performed a comparative analysis of transglycosylation conditions for the synthesis of 2′-deoxyribonucleosides, involving *E. coli* PNP and TP (Table 1). The choice of enzymes of bacterial origin as catalysts was due to their broad substrate specificity, optimal pH in neutral/slightly alkaline media and a sufficiently wide operating temperature range, which allows to perform transglycosylation reactions under mild conditions without noticeable nonspecific cleavage of *N*-glycosidic bond [18,19,20,21,22,23,24]. Then, we analyzed the transglycosylation conditions for enzymatic synthesis of 2′-deoxynucleosides using different sources of sugar moiety and varying nucleoside-donor:Base-acceptor:Phospate-anion (D:B:P*_i_*) ratio (Table 1). For enzymatic transglycosylation, we used a pair of PNP-TP or PNP/TP individually, depending on the sugar moiety source and product of the reaction.

The data listed in Table 1 for 2′-deoxynucleosides were in agreement with the data that we obtained earlier for ribonucleosides [15,25]. In the case, when 2′-deoxyadenosine (dAdo) was obtained from thymidine (Thd), the reaction proceeded with high yield (Table 1, entries 1–4, 87–94% at different ratios D:B:P_i_ according to HPLC data). *E. coli* TP can also catalyze phosphorolysis of both Thd and dUrd and can be recruited for enzymatic synthesis of dUrd from Thd and *vice versa* (Table 1, entries 5–12). When using Thd and Ura as TP’s substrates, dUrd was obtained with 43–58% yields. Comparable substrate properties of initial Thd and the resulting dUrd to TP demanded a three-fold excess of an initial substrate over a base (Ura) in the reaction mixture to increase reaction yield up to 58%. These data suggest that both ribo- and 2′-deoxynucleosides pyrimidine nucleosides are better cleavable carbohydrate residue donors than purine nucleosides. Among purine 2′-deoxynucleosides, 2′-deoxyinosine (dIno) acted as a more productive glycosyl-donor than dAdo; therefore, obtaining dAdo from dIno proceeded with a higher yield (Table 1, entries 21–24, 71–85% according to HPLC data) than the reverse reaction (Table 1, entries 17–20, 45–60%). The formation of dIno from 2′-deoxyguanosine (dGuo) as 2′-deoxyribose donor proceeded with yields comparable with the preparation of dIno from dAdo (62% and 60% correspondingly). Thus, dGuo and dAdo are comparable donors of 2′-deoxyribose residue in enzymatic transglycosylation.

Conducting a reaction with a slight excess of carbohydrate residue donor and a lack of phosphate (D:B:P_i_ 1.5:1:0.25) resulted in the high yields of purine 2′-deoxynucleosides from 2′-deoxypyrimidine donors (Table 1, entry 3, 84% for dAdo from Thd) and good yields of purine 2′-deoxynucleosides from purine donors (Table 1, entries 27, 31, 61% for dAdo from dGuo, 49% for dIno from dGuo). When using purine sugar donors under the lack of phosphate, pyrimidine 2′-deoxynucleosides were obtained with lower yields (Table 1, entry 15, 20% for dUrd from dAdo at D:B:Pi 1.5:1:0.25). 7-Me-dGuo appeared to be a better donor of 2′-deoxyribose moiety than Thd, dAdo, dIno and dGuo due to its practically irreversible phosphorolysis [14,26], leading to high yields of deoxynucleosides under small excess of glycosyl-donor (Table 1, entries 33–51).

To obtain less available deoxyribonucleosides from respective ribonucleosides, we used the ability of *E. coli* PNP to cleave and synthesize both ribonucleosides and 2′-deoxynucleosides [3,4,13,14]. (Figure 1).

Ribonucleosides were chemically synthesized from commercially available reagents according to the previously elaborated protocols: *N*^6^-benzyl-2-aminoadeosnine was obtained from 6-chloro-2-aminopurine riboside by direct substitution of chlorine atom by corresponding modified benzylamine residue [27]. *N*^6^-Furfuryladenosine was obtained starting from *N*^6^-acetyl-2′,3′,5′-tri-O-acetyladenosine by selective alkylation of *N*^6^-acetyl-2′,3′,5′-tri-O-acetyladenosine with furfuryl alcohol in Mitsunobu reaction conditions [28]. In this work, we also adopted this previously optimized procedure for *N*^6^-alkylation of *N*^6^-acetyl-2′,3′,5′-tri-O-acetyladenosine by pentafluorophenyl benzyl bromide in the presence of DBU in dry acetonitrile. *N*^6^-alkylation followed by acetyl deblocking by the treatment with 4 M PrNH_2_/MeOH gave the product *N*^6^-(pentafluorobenzyl)-adenosine as a pure product with 94% total yield (Figure 2).

In ^13^C-NMR spectrum of pentafluoro-substituted substrate **1a**, resonant signals of ^13^C-nuclei corresponding to phenyl group in the form of three wide doublets with and fine line structure and coupling constants ^1^*J*_C-F_ of 250 Hz order of magnitude were present (Figure 2). In the ^19^F-NMR spectrum of pentafluorosubstituted nucleoside **1a**, a complex spin-spin interaction was also observed: a doublet of doublets for ^19^F nuclei in the *ortho*-position of the phenyl substituent with ^3^*J*_F-F_ = 22 Hz, ^4^*J*_F-F_ = 6 Hz, a triplet of doublets for ^19^F nuclei in the *meta*-position with CCC ^3^*J*_F-F_ = 22 Hz, ^4^*J*_F-F_ = 6 Hz and a triplet for ^19^F nuclei in the *para*-position of the phenyl ring with ^3^*J*_F-F_ = 22 Hz. The presence of a pentafluorosubstituted fragment in structure **1a** was also confirmed by the absence of phenyl proton signals in the ^1^H-NMR spectrum (Appendix A).

The proposed enzymatic method of preparation modified 2′-deoxynucleosides consisted of three separate steps; the course of each stage was catalyzed by *E. coli* PNP. These steps were combined to obtain 2′-deoxynucleosides from ribonucleosides. The initial bases *N*^6^-pentafluorobenzyl adenine (*N*^6^-PFBn-Ade, **3a**), *N*^6^-benzyl-2-aminoadenine (*N*^6^-Bn-2-NH_2_-Ade, **3c**) and *N*^6^-furfuryladenine (*N*^6^-Fur-Ade, **3b**) were obtained from the corresponding ribonucleosides under enzymatic arsenolysis or phosphorolysis conditions (Figure 1, step **i**). Enzymatic arsenolysis is based on the cleavage of ribonucleoside in the presence of potassium dihydroorthoarsenate (KH_2_AsO_4_) to a purine base and highly labile α-*D*-ribofuranose-1-arsenate (Rib-1-As), which is irreversibly hydrolyzed, shifting the equilibrium of ribonucleoside cleavage towards the formation of a base [29]. When using enzymatic phosphorolysis conditions in step **i**, the addition of *E. coli* alkaline phosphatase (AP), cleaving dRib-1-P, was required to shift the equilibrium towards forming of a purine base. Poor solubility of purine heterocyclic bases in water and Tris-HCl buffer also leads to an equilibrium shift. To prevent the output of a mixture of ribonucleosides and deoxyribonucleosides during further synthesis, we carried out step **i** in a separate flask. Furthermore, purine bases **3a**–**c** after filtration were introduced into the transglycosylation reaction with 7-Me-dGuo in the presence of potassium dihydroorthophosphate (KH_2_PO_4_) and *E. coli* PNP (Figure 1, steps **ii**–**iii**). In the reaction mixture, 7-Me-dGuo was transformed into dRib-1-P (Figure 1, step **ii**), which then reacted with a purine base (Figure 1, step **iii**). Steps **ii** and **iii** were carried out in the same reaction flask. To increase the solubility of a fluorinated heterocyclic base, the reaction was carried out in a buffer solution containing 10% (*v/v*) DMSO. The concentration of dimethyl sulfoxide in the reaction mixture did not significantly affect the enzymatic activity of NPs, which is consistent with the literature data [16,19]. *N*^6^-Bn-2-NH_2_-Ade and *N*^6^-Fur-Ade were soluble in Tris-HCl buffer up to 1 mM concentration; therefore, transglycosylation reactions involving these purine derivatives were conducted without adding DMSO. When performing transglycosylation reactions in preparative scale at heterocyclic base concentration 1mM, the yield of product **6b**—98% (Figure 3) and **6c** was 100% (Figure 4) according to HPLC and 85–93% for **6b** and **6c** after purification by reverse-phase chromatography on silica gel-C18. Earlier reported pentafluorinated 2′-deoxynucleoside **6a** was isolated with 47% after purification by reverse-phase chromatography on silica gel-C18, most likely due to non-specific sorption on a chromatographic sorbent.

The overall yield of 6a obtained by phosphorolysis of ribonucleoside **1a** with further enzymatic glycosylation of base 3a in the presence of 7-Me-dGuo (**4**) was 28%, which was lower than in previously published procedure including arsenolysis step [25].

The structure of the obtained compounds was confirmed by NMR spectroscopy (see Appendix A for more details). The ^13^C NMR signals of pentafluorinate phenyl residue of **6a** were similar with the spectra of initial substrate 1a (Figure 2).

## 3. Discussion

Transglycosylation reaction catalyzed by nucleoside phosphorylases represents a highly efficient glycosylation method, which is widely used to produce biologically active nucleosides with β-configuration of *N*-glycosidic bond. The first step is phosphorolysis of the initial nucleoside (Nuc1) with the formation of α-*D*-(2-deoxy) ribose-1-phosphate (dRib-1P) and Base1; the second is the synthesis of a new nucleoside (Nuc2) from base (Base2) and dRib-1-P. Depending on the set of starting compounds and nucleoside phosphorylases used, new modified nucleosides (Nuc2) can be obtained. To date, three main approaches to producing biologically active nucleosides (Figure 1) varying different sources of sugar moiety have been developed.

The first is to obtain a dRib-1-P *in situ* using the enzymatic phosphorolysis reaction according to Figure 1 Step 1. Pyrimidine nucleosides are usually utilized as starting nucleosides [3,4,17]. The equilibrium constant of enzymatic phosphorolysis of natural pyrimidine nucleosides in TP’s presence differs markedly upwards from the equilibrium constant of purine deoxynucleosides [3,4,11,13,17], making the use of pyrimidine deoxynucleosides (e.g., Thd) as a source of sugar moiety more preferable than purine ones [13,17]. The data given in Table 1 are in agreement with these conclusions. The data are given in Table 1 also agrees with our previously studied various transglycosylation conditions for the synthesis of pyrimidine and purine ribonucleosides [13,14,15,25]. When we obtained adenosine from uridine, the reaction proceeded with a high yield (77–88% according to HPLC) [25]. When uridine was obtained from adenosine, the reaction yields significantly decreased (22% according to HPLC). Additionally, inosine, as its 2′-deoxy derivative, acted as a more productive carbohydrate residue donor among purine ribonucleosides than adenosine; therefore, the reaction of obtaining adenosine from inosine proceeded with a higher yield (76% according to HPLC data) than the reverse reaction (76% and 53%, respectively) [25].

In the second approach, the hydroiodic salt of 7-methyl-2′-deoxyguanosine (7-Me-dGuo, **4**) is used as substrate, also producing dRib-1-P (**2**, X = P) *in situ* (Figure 1 Step **iii**). 7-Me-dGuo **4** cleaved to dRib-1-P **2** and 7-Me-Gua with nearly quantitative yield due to its practically irreversible phosphorolysis [13,14,26] affording high conversion of purine and pyrimidine glycosyl-acceptor to the corresponding nucleoside, thus increasing their yield compared to non-methylated glycosyl-donor dGuo (Table 1). In addition, that is why 7-Me-dGuo appears to be a better source of 2′-deoxyribose moiety than Thd, whose phosphorolysis is equilibrium.

In the latter approach, the carbohydrate residue source is dRib-1-P directly. The use of dRib-1-P readily reduced the number of components in the composition of the reaction mixture, facilitated the isolation of target compounds and allowed a significant shift in the equilibrium of the glycosylation reaction towards the formation of nucleosides. As we previously reported [25], using Rib-1P as a glycosyl-donor afforded a 98% yield of Ado from Ade (HPLC). The disadvantages of this carbohydrate residue source are still its high cost, high lability and impossibility of long-term storage. At the same time, the replacement Rib-1-P with 7-Me-Guo practically did not reduce the yield of the target nucleoside and led to a comparable result (94% of Ado from Ade) [25].

This work examined the formation of various purine 2′-deoxynucleosides under transglycosylation conditions with different carbohydrate residue sources and varying carbohydrate residue donor and phosphate concentrations. According to the previously reported transglycosylation reaction mathematical model, we can increase the transglycosylation yield when an excess of sugar moiety donor and lack of phosphate is present in the reaction mixture [17]. Concentrations of glycosyl-donor, phosphate and phosphorolysis constants of nucleosides participating in the reaction defined the maximal yield of transglycosylation reaction. In the present research, we proposed different phosphate concentrations, which we used to study and quantify the effect of initial conditions on the yield of 2′-deoxynucleosides obtained by transglycosylation reaction. The transglycosylation reaction was carried out at different ratios of D:B:Pi (Table 1). Lack of phosphate can increase the yield, especially in the preparation of modified 2′-deoxynucleosides, or the yield remains unchanged, as is the case with equimolar phosphate concentrations (Table 1). The obtained experimental results agree with the proposed mathematical model and thermodynamic considerations, as an increased phosphate concentration shifts the equilibrium towards the formation of phosphorolysis products, decreasing the yield of product Nuc2. We used the obtained data for the preparative synthesis of potential biologically active purine-modified 2′-deoxynucleosides.

To optimize the enzymatic synthesis of purine-modified 2′-deoxynucleosides as potential biologically active substances, we studied transglycosylation reactions of synthetic derivatives *N*^6^-(3-trifluoromethylbenzyl)adenine (*N*^6^-TFMBn-Ade) and *N*^6^-pentafluorophenyl methyl adenine (*N*^6^-PFBn-Ade, **3a**), *N*^6^-benzyl-2-aminoadenine (*N*^6^-Bn-2-NH_2_-Ade, **3c**) and a natural plant hormone *N*^6^-furfuryladenine (*N*^6^-Fur-Ade, **3b**), widely known as kinetin. It was earlier shown that the introduction of several fluorine atoms into the phenyl ring of natural compound *N*^6^-benzyladenosine (BAR) increased its antiretroviral activity [30]. The presence of trifluoromethyl group in position 3 of *N*^6^-benzyladenosine substantially increased its antiviral activity, inhibiting human enterovirus EV71 reproduction in nanomolar concentrations and low cytotoxicity for host cells. Therefore, *N*^6^-(3-trifluoromethyl)benzyl-2′-deoxyadenosine can be considered a prospective compound with high antiviral potential. (Pentafluorobenzyl)-2′-deoxyadenosine (**6a**) represents a valuable synthetic substrate for various chemical modifications of phenyl’s position 4 by substitution of fluorine atom with nucleophilic reagents (alkylamine, morpholine, piperazine, piperidine, pyrrolidine and others) [31,32,33] and further modulation of antiviral activity and cytotoxicity profiles of the resulting nucleoside derivatives depending on the structure of the introduced substituent. *N*^6^-Benzyl-2-aminopurine-2′-deoxyadenosine is also a substrate for preparing a set of biologically active compounds by chemical modification of aminogroup at purine C-2 position [34]. *N*^6^-furfuryladenine and *N*^6^-benzyladenine participate in the regulation of growth and development of plant cells by interaction with specific cytokinin receptors [35,36]. Glycosylated cytokinins do not possess biological activity and can be considered depot-forms of active hormones in plants and can exhibit biological activity on animal cells. Therefore, the application of modified plant hormones 2′-deoxyribosylkinetin and *N*^6^-benzyl-2-aminopurine-2′-deoxyadenosine may be helpful for the study of glycosylation role in metabolic activation and deactivation of purine derivatives in plant and animal cells.

To optimize conditions of preparative synthesis, we performed several analytical reactions catalyzed by *E. coli* PNP with 7-Me-dGuo as a glycosyl-donor and modified purine bases *N*^6^-TFMBn-Ade, *N*^6^-PFBn-Ade, *N*^6^-Bn-2-NH_2_-Ade, *N*^6^-Fur-Ade as glycosyl-acceptors (Table 1). HPLC-analysis was used to analyze reaction mixtures; the yields were taken as 100% when no detectable peak of the initial base on HPLC was observed. In analytical experiments, a correlation between yield and phosphate concentration was observed. The highest yields, tending to 100%, were achieved with the ratio 7-Me-dGuo:Base:phosphate—1.5:1:0.25 (Table 1). When ratio 7-Me-dGuo:Base:phosphate was 1.5:1:1, the yields of products decreased by 4–11% (Table 1, entries 42, 45, 89% for *N*^6^-PFBn-dAdo, 96% for *N*^6^-TFMBn-dAdo at 7-Me-dGuo:Base:phosphate) as increased phosphate concentration shifts the equilibrium towards phosphorolysis products, base and dRib-1P. These data agree with the mathematical model and equilibrium analysis: an excess of phosphate in a mixture shifts the equilibrium of a reaction towards the formation of dRib-1-P in the case of both nucleoside-donor and nucleoside-product.

When preparing 2′-deoxynucleosides from ribonucleosides, *N*-glycosidic bond in nucleosides (donors of heterocyclic base) can be cleaved by two various enzymatic methods: enzymatic phosphorolysis in the presence of *E. coli* purine nucleoside phosphorylase and *E. coli* alkaline phosphatase [28] and by enzymatic arsenolysis in the presence of *E. coli* purine nucleoside phosphorylase and potassium dihydroorthoarsenate (KH_2_AsO_4_) [29,37,38]. Synthesis of **3a**–**c** by enzymatic arsenolysis and **3b** by enzymatic phosphorolysis was described earlier [28,37,38]. In this work, **3a** was obtained in enzymatic phosphorolysis conditions with lower yield (60%) than under arsenolysis conditions [25]. The advantage of the enzymatic arsenolysis compared to the enzymatic phosphorolysis is the possibility of carrying out reactions in a short time (16 h) using only one enzyme PNP and a small amount of arsenate. In contrast, the phosphorolysis reaction requires an excess of phosphate and additional use of alkaline phosphatase and proceeds over several days [28,37]. As a result, the products were formed with lower yields than in enzymatic arsenolysis reactions. Therefore, phosphorolysis of ribonucleosides in the presence of alkaline phosphatase seems to be an eco-friendlier method if performed in large-scale synthesis. At the same time, arsenolysis, involving only one enzyme, appears to be more cost-effective.

Our experimental data showed that lowering the phosphate concentration down to 0.25 equivalent does not significantly affect the enzymatic transglycosylation reaction rate. At the same time, a low phosphate concentration simplified the process of purification and isolation of the target 2′-deoxynucleoside from the reaction mixture. The maximal yield can be achieved under small excess of 7-Me-dGuo and a 4-fold lack of phosphate at a 1.5:1:0.25 (D:B:Pi, Table 1) ratio. We believe that this ratio is optimal for obtaining nucleosides by enzymatic transglycosylation, especially modified ones. Additionally, it is possible to increase the yield just by using less phosphate instead of adding the excess of costly nucleoside-donor.

## 4. Materials and Methods

The solvents and materials were reagent grade and were used without additional purification. Column chromatography was performed on silica gel (Kieselgel 60 Merck KGaA, Darmstadt, Germany, 0.063–0.200 mm). TLC was performed on an Alugram SIL G/UV254 (Macherey-Nagel, Düren, Germany) with UV visualization. Ribonucleoside derivatives (**1a**, **1c**) were obtained by reaction of 6-chloropurine riboside derivatives with benzylamines [27], while compound (**1b**) was obtained by alkylation of *N*^6^-acetyl-2′,3′,5′-tri-O-acetyladenosine by furfuryl alcohol under Misunobu conditions according to procedure [28]. Purine bases can be obtained by enzymatic arsenolysis (**3a**–**c**) or enzymatic phosphorolysis (**3b**) of corresponding ribonucleosides according to the previously described procedures [28,37,38]. 7-Methyl-2′-deoxyguanosine (**4**) was obtained as hydroiodic salt by a previously optimized method [13,39]. Compound **6a** can be obtained according to procedure [25]. The ^1^H,^13^C, ^19^F (with complete proton decoupling) NMR spectra were recorded on Bruker AVANCE II 300 (Karlsruhe, Germany) instrument at 293 K. The ^1^H-NMR-spectra were recorded at 300.1 MHz, the ^13^C-NMR-spectra at 75.5 MHz and the ^19^F-NMR-spectra at 282.4 MHz. Chemical shifts in ppm were measured relative to the residual solvent signals as internal standards (DMSO-*d*_6_, 1H: 2.50 ppm, ^13^C: 39.5 ppm). Spin-spin coupling constants (J) are given in hertz (*Hz)*. The following abbreviations were used in description of NMR spectra: s—singlet, br.s—broad singlet, d—doublet, dd—doublet of doublets, ddd—doublet of doublets of doublets, t—triplet, dt—doublet of triplets, m—multiplet. HPLC analysis was performed using Akvilon (Moscow, Russia) HPLC system (2 × Stayer pumps (2nd series), Stayer MS16 dynamic mixer and Stayer 104 M UV-Vis detector. Enzymatic reactions were assessed by HPLC analysis (elution conditions and details can be found in the supporting materials). HPLC analysis was performed for 20 µL analytical volume. The analysis was performed on the following HPLC columns and conditions:

(a) 4.6 mm × 150 mm column (5 µm, Cosmosil 5C18-MS-II, approx. 120 Å, Part No 38019-81, Nacalai Tesque, Inc. (Kyoto, Japan)) equipped with EC security guard (4.0 mm × 3 mm, 5 µm, C_18_ Part No AJ0-4287, Phenomenex (Torrance, CA, USA)) for enzymatic transglycosilation reactions with Thd and Ade (Ura), dUrd and 5-Me-Ura, dAdo and Ura, 7-Me-dGuo and Ura (Hyp, *N*^6^-Bn-Gua). HPLC analysis was run in a linear acetonitrile gradient in 0.06% (*v/v*) TFA/deionized water from 2 to 12% in 10 min (flushing with 12–80% acetonitrile-0.06% TFA/deionized water in 10–10.1 min, then 80–2% in 10.1–10.8 min) at a flow rate of 1 mL/min with UV detection at a wavelength of 260 nm for Thd and Ade (Ura), dUrd and 5-Me-Ura, dAdo and Ura, 7-Me-dGuo and Ura (Hyp). For the reaction 7-Me-dGuo and *N*^6^-Bn-Gua, HPLC analysis conditions were run in a linear acetonitrile gradient in 0.06% (*v/v*) TFA/deionized water from 2 to 30% in 15 min (flushing with 12–80% acetonitrile-0.06% TFA/deionized water in 15–15.1 min, then 80–2% in 15.1–15.8 min) at a flow rate of 1 mL/min with UV detection at 283 nm, and an injection volume of 20 μL.

(b) 4.6 mm × 250 mm column (5 µm, Cosmosil 5CN-MS, 12 approx. 300 Å, Part No 38236-31, Nacalai Tesque, Inc. (Kyoto, Japan)) equipped with Rheodyne inline filter (2 µm) for enzymatic transglycosilation reactions with dAdo and Hyp, dIno and Ade, dGuo and Ade (Hyp). HPLC analysis was run in a linear acetonitrile gradient in 10 mM NaOAc/deionized water from 2 to 12% in 10 min (flushing with 12–80% acetonitrile-10 mM NaOAc/deionized water in 10–10.1 min, then 80–2% in 10.1–10.8 min) at a flow rate of 1 mL/min with UV detection at a wavelength of 260 nm, and an injection volume of 20 μL.

Detection of UV-light absorption by nucleoside products and impurities during reverse-phase chromatographic purification was performed on two-channel UV-detector TOY18DAD400H (ECOM spol. S r.o., Chrastany u Prahy, Czech Republic) equipped with two simultaneously operating channels: channel A—detection at a wavelength of 267 nm, channel B—detection at a wavelength of 283 nm.

### 4.1. N^6^-(2,3,4,5,6-Pentafluorobenzyl)-Adenosine (1a)

A mixture of *N*^6^-acetyl-2′,3′,5′-tri-*O*-acetyl adenosine (300 mg, 0.68 mmol), 2,3,4,5,6-pentafluorobenzyl bromide (0.309 mL, 2.049 mmol) and DBU (0.306 mL, 2.049 mmol) in dry acetonitrile (7 mL) was kept at ambient temperature for 48 h. The reaction mixture was neutralized with 0.1 M HCl (0.04 mL, 0.41 mmol) to pH = 7 and evaporated in vacuum. The residual syrup was diluted with ethyl acetate (15 mL) and washed successively with brine (2 × 10 mL), 10% aqueous sodium bicarbonate (20 mL) and water (2 × 10 mL). The organic layer was separated, dried over anhydrous sodium sulfate and evaporated in vacuum. The residue was applied to column chromatography. The product was eluted with methylene chloride:ethanol—98:2. Purified triacetyl compound was dissolved in 4M PrNH_2_ in MeOH solution (34 mmol) and was left for 24 h, after which the mixture was evaporated and the residue was applied to column chromatography. The column was washed with methylene chloride:ethanol—95:5 and then eluted with methylene chloride:ethanol—90:10 to give *N*^6^-(2,3,4,5,6-pentafluorobenzyl) adenosine as a white powder. Yield 287 mg (94% for two steps). R*_f_* =0.26 (CH_2_Cl_2_:EtOH—97:3). M.p. 180–183°C. ^1^H NMR (400 MHz, DMSO-*d*_6_): δ = 3.55 (ddd, 1H, *J*_5′b,5′a_ = −12.0, *J*_5′b,4′_ = 3.4, *J*_5′b,OH_ = 7.0, H5′b), 3.67 (ddd, 1H, *J*_5′a-5′b_ = −12.0, *J*_5′a-4′_ = 3.4, *J*_5′a-OH_ = 4.7, H5′a), 3.96 (q, 1H, *J*_4′,5′a_ = *J*_4′,5′b_ = *J*_4′,3´_ = 3.4, H4′), 4.14 (td, 1H, *J*_3′,2′_ = *J*_3′,OH_ = 4.7, *J*_3′,4′_ = 3.4, H3′), 4.59 (td,1H, *J*_2′-1′_ = *J*_2′-OH_ = 6.2, *J*_2′-3′_ = 4.7, H2′), 4.82 (br s, 2H, CH_2_), 5.17 (d, 1H, *J*_OH-3′_ = 4.7, 3′OH), 5.29 (dd, 1H, *J*_OH-5′b_ = 7.0, *J*_OH-5′a_ = 4.7, 5′OH), 5.43 (d, 1H, *J*_OH-2′_ = 6.2, 2′OH), 5.89 (d, 1H, *J*_1′-2′_ = 6.2, H1), 8.25 (s, 1H, H-2), 8.39 (br s, 2H, H-8, NH). ^13^C NMR (100 MHz, DMSO-*d*_6_): δ = 32.21 (NCH_2_-), 61.55 (C5′), 70.54 (C3′), 73.55 (C2′), 85.81 (C4′), 87.86 (C1′), 112.98 (C1-Ph), 119.73 (C5), 136.81 (d, ^1^*J*_C-F_ = 246.1), 132.48 (d, ^1^*J*_C-F_ = 166.1), 147.79 (d, ^1^*J*_C-F_ = 154.7) (C2-Ph,C3-Ph,C4-Ph,C5-Ph,C6-Ph), 140.10 (C8), 148.49 (C4 is overlapping with Ph), 152.15 (C2), 153.90 (C6). HRMS: *м/z* [C_17_H_14_F_5_N_5_O_4_+H]^+^ calculated 448.1038, found 448.1039.

### 4.2. General Procedure for the Preparation of Purine Bases by Phosphorolysis of Ribonucleosides

To the suspension of ribonucleoside **1a** or **1b** (62 and 48 mg, respectively, 0.14 mmol) in KH_2_PO_4_ buffer (50 mM, pH 7.5, 10 mL), *E. coli* PNP (1.92 U) was added. The mixture was neatly agitated at 30 °C. On the next day, *E. coli* alkaline phosphatase and (5.2 U) and MgSO_4_ (18 mg) were added and the reaction mixture was left to stay for 4 days at ambient temperature. The mixture was then concentrated in vacuum to a volume of ca. 5 mL and kept at 0 °C overnight. The precipitate was centrifuged, washed with cold deionized water (5 °C, 4 × 2.5 mL) and dried in vacuum desiccator over P_2_O_5_ to give product 3 in 52–60% yield. NMR spectra of 3a and 3b correspond in all details to what has been previously reported [25,28].

### 4.3. Analytical Experiments

The reactions were carried out in 1.5 mL plastic tubes (Eppendorf, Hamburg, Germany), the volume of the reaction mixture was 1 mL in 50 mM Tris-HCl (pH 7.5) buffer, substrate concentrations Donor:Base:P_i_ in the range 0.01 mM-1mM. The reactions were catalyzed by addition of 1 U *E. Coli* PNP, 1 U *E. Coli* TP. The reactions were analyzed on Akvilon HPLC system in the following conditions: linear gradient 2–12% CH_3_CN in 0.06% aqueous TFA solution in 10 min, column 4.6 mm × 150 mm Cosmosil 5C18-MS-II (Table 1, entries 1–16, 33–40); 2–30% CH_3_CN in 0.06% aqueous TFA solution in 15 min, column Cosmosil 5C18-MS-II (Table 1, entries 48–50); 2–12% CH_3_CN in 10 mM aqueous sodium acetate solution in 10 min, 4.6 mm × 250 mm column Cosmosil 5CN-MS (Table 1, entries 17–32); 2–60% CH_3_CN in 10 mM aqueous sodium acetate solution in 25 min, column 4.6 mm × 250 mm Nucleosil C18 (Table 1, entries 41–47).

### 4.4. N^6^-Furfuryl-2′-Deoxyadenosine (**6b**)

*N*^6^-furfuryladenine (**1b**) (29 mg, 0.1386 mmol) was dissolved in 140 mL of 50 mM Tris-HCl buffer (pH 7.5) with addition of 0.7 mL of 50 mM sodium phosphate buffer (0.035 mmol Pi, 0.7 mL). 0.98 U of *E. coli* PNP (10 µL, 1.0 mg/mL, 98 U/mL, Sigma) was then slowly added at ambient temperature to a mixture of 1b and 7-methyl-2′-deoxyguanosine (85.6 mg, 0.21 mmol). The mixture was left to stay at ambient temperature for 24 h under gentle stirring. The precipitate of 7-methylguanine (7-Me-Gua) was filtered through nylon Phenomenex membrane (diam. 47 mm, pore diam. 0.2 µm). The transparent liquid filtrate was evaporated in vacuum to a volume *ca.* 7 mL, diluted with deionized water and applied on a chromatographic column containing C_18_-modified silica-gel (100 mL) as a reverse-phase. The column was washed with H_2_O:ethanol mixture with ethanol gradient from 0 to 20%. The product was eluted in H_2_O:ethanol = 80:20 (*v*/*v*). Fractions containing the product were collected, evaporated in vacuum and dried on a vacuum pump to yield 39 mg (85%) of 7b as a syrup. ^1^H-NMR (400 MHz, CD_3_OD): δ = 8.24 (brs, 2H, H8, H2 Ade), 7.41 (dd, 1H, ^3^*J* = 1.8, ^4^*J* = 0.9, H-5_Fur_), 6.42 (dd, 1H, *J*_1′,2′a_ = 8.0, *J*_1′,2′b_ = 6.0, H-1′), 6.32 (dd, 2H, ^3^*J* = 1.8, ^3^*J* = 3.2, H-4_Fur_), 6.29 (dd, 2H, ^3^*J* = 3.2, ^3^*J* = 0.9, H-3_Fur_), 4.83 (br s, 1H, NH), 4.80 (NCH_2_), 4.58 (ddd, 1H, *J*_3′2′a_ = 2.1, *J*_3′2′a_ = 5.7, *J*_3′4′_ = 2.7, H-3′), 4.07 (ddd, 1H, *J*_4′3′_ = 2.7, *J*_4′5′a_ = 2.9, *J*_4′5′b_ = 3.4, H-4′), 3.85 (dd, 1H, *J*_5′a5′b_ = −12.3, *J*_5′a4′_ = 2.9, H-5′a), 3.73 (dd, 1H, *J*_5′a5′b_ = −12.3, *J*_5′b4′_ = 3.4, H-5′b), 2.80 (ddd, 1H, *J*_2′a2′b_ = −13.6, *J*_2′a1′_ = 8.0, *J*_2′a3′_ = 5.8, H-2′a), 2.40 (ddd, 1H, *J*_2′b2′a_ = −13.6, *J*_2′b1′_ = 6.0, *J*_2′b3′_ = 2.7, H-2′b). ^13^C-NMR (150 MHz, CD_3_OD): 155.89 (C-6), 153.42 (Fur), 149.38 (C-4), 143.32 (Fur), 141.17 (C-8), 121.29 (C-5), 111.35 (Fur), 108.19 (Fur), 89.88 (C-1′), 87.13 (C-3′), 73.04 (C-4′), 63.64 (C-5′), 41.55 (C-2′), 38.37 (N*C*H_2_).

### 4.5. N^6^-Benzyl-2-Amino-2ʹ-Deoxyadenosine (**6c**)

In total, 0.48 U of *E. coli* PNP (5 µL, 1.0 mg/mL, 98 U/mL, Sigma) was slowly added at ambient temperature to a mixture of 7-methyl-2′-deoxyguanosine (20.2 mg, 0.049 mmol), *N*^6^-benzyl-2-aminoadenine (8.0 mg, 0.033 mmol) and KH_2_PO_4_ (0.00825 mmol, 0.165 mL 50 mM sodium-phosphate buffer) in 33 mL of 50 mM Tris-HCl-buffer (pH 7.5). The mixture was left to stay at ambient temperature for 2 h under gentle stirring and was then left to stay at 0 °C overnight. The precipitate of 7-Me-Gua was filtered through a Phenomenex membrane (diam. 47 mm, pore diam. 0.2 µm). The transparent liquid filtrate was evaporated in vacuum to a volume of *ca.* 10 mL, filtered from mechanical impurities through a cellulose membrane filter and divided into 20 parts, 500 μL each, which were subjected to HPLC purification. HPLC purification was run using the Akvilon (Moscow, Russia) HPLC system (2 × Stayer pumps (2nd series), Stayer MS16 dynamic mixer and Stayer 104M UV-Vis detector) and semi-prep 10.0 mm × 250 mm column (5 µm, Cosmosil 5C18-MS-II, approx. 120 Å, Part No 38023-11, Nacalai Tesque, Inc. (Kyoto, Japan)) equipped with EC security guard (10.0 mm × 10 mm, C_18_ Part No AJ0-7221, Phenomenex (Torrance, CA, USA)) HPLC purification was perform in a linear acetonitrile gradient in deionized water from 2 to 30% in 15 min (flushing with 30–80% acetonitrile-deionized water in 15–15.1 min, then 80–2% in 15.1–15.8 min) at a flow rate of 3 mL/min with UV detection at a wavelength of 283 nm, and an injection volume of 500 μL. he product was eluted after flushing step in gradient program. Fractions containing the product were collected, evaporated in vacuum co-evaporated with absolute methanol and dried on a vacuum pump for 1 h to yield 11 mg (93%) of 7 d as a white foam. ^1^H-NMR (400 MHz, CD_3_OD): δ = 7.87 (s, 1H, H8-Pur), 7.36 (dd, 2H, ^3^*J* = 7.2, ^4^*J* = 1.7, *H*-o-Ph), 7.30 (dd, 2H, ^3^*J_o_* = 7.2, ^3^*J_p_* = 7.3, *H*-m-Ph), 7.22 tt (1H, ^3^*J_m_* = 7.3, ^4^*J* = 1.7, *H*-p-Ph), 6.26 (dd, 1H, *J*_1′2′a_ = 8.5, *J*_1′2′b_ = 5.9, H-1′), 4.74 (br s, 2H, CH_2_), 4.55 (ddd, 1H, *J*_3′2′a_ = 2.1, *J*_3′2′a_ = 5.7, *J*_3′4′_ = 2.7, H-3′), 4.05 (ddd, 1H, *J*_4′5′a_ = *J*_4′3′_ = 2.7, *J*_4′5′b_ = 2.9, H-4′), 3.85 (dd, 1H, *J*_5′a5′b_ = −12.3, *J*_5′a4′_ = 2.7, H-5′a), 3.73 (dd, 1H, *J*_5′a5′b_ = −12.3, *J*_5′b4′_ = 2.9, H-5′b), 2.80 (ddd, 1H, *J*_2′a2′b_ = −13.9, *J*_2′a1′_ = 8.5, *J*_2′a3′_ = 5.7, H-2′a), 2.30 (ddd, 1H, *J*_2′b2′a_ = −13.9, *J*_2′b1′_ = 5.9, *J*_2′b3′_ = 2.1, H-2′b). ^13^C-NMR (150 MHz, CD_3_OD): 161.59 (C-4), 156.47 (C-6), 140.57 (Ph), 138.18 (Ph), 129.49 (Ph), 128.55 (Ph), 128.12 (Ph), 115.45 (C-5), 89.89 (C-1′), 87.27 (C-3′), 73.43 (C-4′), 63.97 (C-5′), 44.91 (CH_2_), 41.19 (C-2′).

## 5. Conclusions

In the course of this work, we performed a comparative analysis of transglycosylation conditions for the preparation of 2′-deoxyribonucleosides in the presence of *E. coli* nucleoside phosphorylases. It was shown that maximal yields of 2′-deoxyribonucleosides, especially modified, can be achieved under small excess of glycosyl-donor (Thd, 7-Me-dGuo) and a 4-fold lack of phosphate. The optimized conditions were then applied for preparative enzymatic synthesis of three biologically active nucleosides, *N*^6^-(pentafluorobenzyl)-, *N*^6^-furfuryl-and *N*^6^-benzyl-2-amino-2′-deoxyadenosine, starting from the corresponding ribonucleosides. The proposed three-step synthetic procedure consisted of three enzymatic steps. Ribonucleosides were introduced into phosphorolysis reaction catalyzed by *E. coli* PNP and *E. coli* AP or arsenolysis reaction catalyzed by *E. coli* PNP with the formation of corresponding purine bases. The bases were then subjected to transglycosylation using 7-Me-dGuo as a glycosyl-donor in the presence of *E. coli* PNP and a lack of phosphate (7MedGuo:B:Pi ratio 1.5:1:0.25) and further isolation by chromatographic purification to obtain pure target compounds with good and high yields (47–93%). The antiviral and cytokinin activity of the synthesized compounds will be examined further in our future work.

## Data Availability

Not applicable.

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
