# Peer review of "Comparative Analysis of Enzymatic Transglycosylation Using E. coli Nucleoside Phosphorylases: A Synthetic Concept for the Preparation of Purine Modified 2′-Deoxyribonucleosides from Ribonucleosides"

_ijms, 2022, doi:10.3390/ijms23052795_

Round 1
Reviewer 1 Report
The manuscript by Drenichev et al. presents the optimization of enzymatic transglycosylation reaction conditions and the application of the optimized conditions to synthesize bioactive nucleosides.
Many modified nucleosides are drugs or drug candidates, especially for antiviral agents. Compared to traditional chemical synthesis, enzymatic synthesis have better regio- and stereo- selectivity and is more environmentally friendly. However, the related enzymes in these reactions catalyze reversible reactions, and more than one nucleoside phosphorylases are usually used in one reaction, leading to complex components in the reactions. Hence, optimizing the reaction to accumulate the target compound is extremely important.
The authors previously proposed a mathematical model of the transglycosylation process. Here, they designed reactions conditions based on their model, found the best component ratios, and synthesized important modified nucleosides with excellent yields. I will recommend acceptance after the authors address the following minor points:
- I was a little bit confused about the name of the enzymes. Does "nucleoside phosphorylases E. coli" mean nucleoside phosphorylases from E. coli? If yes, it might be better to say " E. coli nucleoside phosphorylases". The authors also use "E. coli PNP and TP" (Line 93), "TP E. coli" (Line 111), and so on. Please unify the names.
- In Table 1. What is "HPLC defined base convertion"? And why are these data lacking for most entries?
- The compound numbering is confusing. The author should unify the numbers in Scheme 1 and Figure3/4: use a unique number for each compound throughout the paper.
- In the ligand of Figure 3, N6-Fur-dAdo should be 6b.
Author Response
Please see the attachment with the response.
I'm not quite sure should I send the attached revised manuscript (with the "Track Changes" option "On") now or later when the options to upload a new version are available.
Yours Faithfully,
Dr Cyril Alexeev

Reviewer 2 Report
This manuscript entitled "Comparative analysis of enzymatic transglycosylation using nucleoside phosphorylases E. coli: a synthetic concept for the 3 preparation of purine modified 2ʹ-deoxyribonucleosides from ribonucleosides.", reports interesting and novel data.
The authors of this manuscript describe accurately all used methods; all figures, reactions schemes and tables are the approptiate ones. Additionally, authors provide to the readers, by means of the supplementary text, all the necessary information on their experimental work.
Although the text is well followed, however there are some small syntax errors which should be taken into account by authors and be corrected.
Overall, I suggest the publication of this manuscript in the International Journal of Molecular Sciences, according to the abovementioned comments.
Author Response

(The authors gave the same response as above.)
